# Extracts of Digested Berries Increase the Survival of *Saccharomyces cerevisiae* during H_2_O_2_ Induced Oxidative Stress

**DOI:** 10.3390/molecules26041057

**Published:** 2021-02-18

**Authors:** Gabriel Oliveira, Nataša Radovanovic, Maria Cecilia do Nascimento Nunes, Rikard Fristedt, Marie Alminger, Thomas Andlid

**Affiliations:** 1Department of Biology and Biological Engineering, Food and Nutrition Science, Chalmers University of Technology, Kemivägen 10, SE-412 96 Gothenburg, Sweden; biohits@gmail.com (G.O.); natasabb.radovanovic@gmail.com (N.R.); rikfri@chalmers.se (R.F.); marie.alminger@chalmers.se (M.A.); 2Food Quality Laboratory, Department of Cell Biology, Microbiology and Molecular Biology, University of South Florida, Tampa, FL 33620, USA; mariacecilia@usf.edu

**Keywords:** polyphenols, anthocyanins, in vitro digestion, oxidative stress, improved survival, yeast, stress bioassay, berry fruits

## Abstract

Many studies suggest anthocyanins may prevent the development of several diseases. However, anthocyanin bioactivity against cellular stress is not fully understood. This study aimed to evaluate the protective effect of berry anthocyanins on stressed cells using *Saccharomyces cerevisiae*. The impact of in vitro gastrointestinal digestion on anthocyanin profiles was also assessed. Bilberry and blackcurrant had higher anthocyanin levels than raspberry and strawberry, but digestion reduced the detected anthocyanins by approximately 90%. Yeast cells with and without digested or nondigested anthocyanin extracts were exposed to H_2_O_2_ and examined for survival. In the presence of anthocyanins, particularly from digested strawberry, a significant increase in cell survival was observed, suggesting that the type and levels of anthocyanins are important factors, but they also need to undergo gastrointestinal (GI) structural modifications to induce cell defence. Results also showed that cells need to be exposed to anthocyanins before the stress was applied, suggesting induction of a cellular defence system by anthocyanins or their derivatives rather than by a direct antioxidative effect on H_2_O_2_. Overall, data showed that exposure of severely stressed yeast cells to digested berry extracts improved cell survival. The findings also showed the importance of considering gastrointestinal digestion when evaluating anthocyanins’ biological activity.

## 1. Introduction

Berry fruits such as bilberry (*Vaccinium myrtillus*), blackcurrant (*Ribes nigrum*), raspberry (*Rubus idaeus*) and strawberry (*Fragaria x ananassa*) are rich natural sources of phenolic compounds, particularly anthocyanins [1,2]. Anthocyanins are mostly found in their glycosylated form in the vacuoles of plants. Glucose, galactose or arabinose are the most common sugar moieties linked to the aglycon (anthocyanidin). The glycosylation contributes to pigment stability and increases the solubility inside vacuoles in the plant cells where the low pH and vacuolar conditions prevent anthocyanin oxidation [3,4,5,6].

Several studies have shown that anthocyanins may help protect against many chronic diseases [7,8,9,10,11], and accumulated evidence suggests that anthocyanins may induce intracellular protection against oxidative stress [12,13,14,15]. For instance, increasing mRNA levels of heme oxygenase 1 (HO-1) and consequently protecting retinal cells from diabetes-induced oxidative stress [15]. It has also been suggested that differences in polyphenol composition between berry species may account for differences in bioactivity against a range of disorders [16]. However, anthocyanins are sensitive to physicochemical conditions encountered during gastrointestinal digestion, and may also be modified during absorption and metabolism, leading to structural changes into anthocyanin derivatives with different biochemical properties. After berry ingestion, the low pH in the stomach may contribute to anthocyanin structural form stability. Still, upon reaching the small intestine, where pH is approximately neutral, anthocyanins become unstable and are transformed mostly into different monomeric phenolic acids and aldehydes [17]. Moreover, intestinal enzymes and microbiota can further metabolize anthocyanins, explaining their low bioaccessibility [17]. For instance, the fraction of recovered anthocyanins has been reported to be less than 7% after in vitro gastrointestinal digestion [18]. Consequently, the concentration of intact anthocyanins, in their original form as present in the plant, reaching the target tissues may be very low. However, other possibilities for the observed protection by berry consumption are possible, for example, early absorption in the stomach of polyphenols, including anthocyanins, and/or the likelihood of berry anthocyanins not being the main compounds responsible for the effect but rather the derivatives resulting from biophysiochemical transformation.

There are studies demonstrating absorption of anthocyanins in the stomach of rats [19]. It has been suggested that absorption of anthocyanins in the stomach also happens in humans, based on observations of their early presence in the blood after ingestion [20]. A few published studies have shown the impact of anthocyanins and polyphenol extracts, including derivatives resulting from gastrointestinal (GI) digestion on living cells. For example, Garcia et al. [21] showed that in vitro digested raspberry extracts at low concentrations exhibit neuronal protection against H_2_O_2_-induced oxidative stress and lipopolysaccharide (LPS)-induced inflammation. In a recent study, Garcia et al. [22] further suggested that digested raspberry extracts enriched in ellagitannins and ellagic acid derivatives are promising compounds in alleviating neuroinflammation. Further studies are needed to understand how anthocyanins and (poly)phenols in general exert beneficial effects, what concentrations are necessary, and which are the most biologically active forms of these compounds.

Bioactivity must be studied using living organisms, such as the much-used model *Saccharomyces cerevisiae*. Baker’s yeast *S. cerevisiae* is a well-characterized eukaryotic organism providing a relevant biological model for studying cellular pathologies since its genome is significantly similar to that of humans [23]. Moreover, the use of *S. cerevisiae* as a cellular model is less expensive and avoids ethical considerations compared to animal models. The yeast and mammalian oxidative stress responses are similar. More than 25% of the genes related to aging and human-degenerative pathologies, such as cancer and Parkinson’s disease, have close orthologs in yeast [24]. Potentially, a yeast model for anthocyanin response would, together with simulated GI-digestion and advanced analysis, constitute an efficient and relevant approach to investigate the impact of gastrointestinal conditions on bioactivity of berry extracts on live cells.

The present study aimed to evaluate the effect of anthocyanin extracts from berries on the survival of *S. cerevisiae* exposed to H_2_O_2_-induced oxidative stress. In vitro GI-digested and nondigested berry extracts were used to address whether anthocyanin structural modifications and conversion into derivatives in the gut affect their bioactivity and cell protective potential.

## 2. Results

### 2.1. Anthocyanin Profiles of Nondigested and Digested Berries

The type and amount of glycosylated and nonglycosylated anthocyanin varied significantly between berry fruit (Table 1). The chromatograms are shown in the Supplementary Material (SM, Appendix A). Bilberry had the highest levels of anthocyanins (11.78 mg g^−1^) followed by blackcurrant (5.03 mg g^−1^). Raspberry and strawberry had significantly lower levels of anthocyanins compared with the other berries (0.89 and 0.74 mg g^−1^, respectively). Moreover, bilberry and blackcurrant showed a much higher diversity of anthocyanin compounds (15 and 12 identified anthocyanins, respectively) compared to raspberry and strawberry (five and four compounds identified, respectively). In bilberry, the most abundant compounds identified (≥1 mg g^−1^) were cyanidin-3*-O-*arabinose, cyanidin-3*-O-*glucoside, delphinidin-3*-O-*galactoside, delphinidin-3*-O-*glucoside, delphinidin-3*-O-*rutinoside and malvidin-3*-O-*glucoside, and the aglycone pelargonidin. Although in blackcurrant, these compounds were present at lower levels than in bilberry, the amounts of cyanidin-3*-O-*galactoside and cyanidin-3*-O-*rutinoside were higher in blackcurrant. In raspberry and strawberry, the compounds identified had much lower levels compared to the other berries. In raspberry, delphinidin-3*-O-*rutinoside was the primary compound identified, and in strawberry cyanidin-3*-O-*rutinoside was identified as the most abundant anthocyanin.

Depending on the type of berry fruit, GI digestion contributed to a significant decrease in each compound (83 to 98%, depending on the compound). On average, a reduction in the anthocyanin levels after digestion was higher in bilberry and raspberry (95 and 91% decrease, respectively) compared to blackcurrant and strawberry (89 and 86%, respectively). In some cases, few anthocyanins were not detected in the digested fruit. For example, in bilberry, the aglycones cyanidin, malvidin and peonidin present in the nondigested fruit were not detectable after GI digestion. Similarly, in blackcurrant, cyanidin-3*-O-*arabinose, malvidin-3*-O-*glucoside and peonidin were not detectable after berry digestion. On the other hand, in blackcurrant, raspberry and strawberry, some compounds not detectable in the nondigested fruit were detected after GI digestion. These compounds were pelargoidin-3*-O-*glucoside, delphinidin and malvidin in blackcurrant, delphinidin-3*-O-*rutinosoide in raspberry and cyanidin in strawberry.

Although the levels of anthocyanins were higher in bilberry than strawberry, the decrease after digestion was higher in bilberry than in strawberry. In bilberry, several compounds were not detected after digestion. In strawberry, all compounds present before digestion were also detected after digestion, but one new compound (cyanidin aglycone), was also detected in digested strawberry.

### 2.2. Determination of Respiratory State and Optimal H_2_O_2_ Concentration for Oxidative Stress Assay

The Crabtree-positive yeast *Saccharomyces cerevisiae* SKQ2n displays a typical diauxic growth pattern (two exponential phases with different catabolism and growth rates) when cultured in a synthetic medium with one single carbon energy source (Figure 1). It is possible to identify these patterns by monitoring growth, allowing cells in different physiological states to be reproducibly studied. This allows for selecting mainly fermenting cells, as in the respiro-fermentative phase (RF in Figure 1), or completely respiring cells (R in Figure 1) as in the respiratory phase in which the yeast respires earlier-formed ethanol. Since the intention was to assess oxidative stress resistance, we chose to collect cells in a state of oxidative catabolism (i.e., when the yeast cells defence system for reactive oxygen species is active). Therefore, we harvested cells after nine hours of growth, which is in the middle of the respiratory phase. All subsequent stress experiments with berry extracts in this study were performed with cells harvested at this time-point (nine hours; midrespiratory phase).

Figure 2 shows the survival rate of *S. cerevisiae* SKQ2n after 10 min of exposure to H_2_O_2_ in a concentration range of 0.1–1.4 M. The fraction of surviving cells decreased with increasing H_2_O_2_ concentrations up to 1.4 M, at which the survival was around 0%. An H_2_O_2_ concentration of 0.45 M, which yields survival of approximately 25% after 10 min, was selected for all subsequent studies since it allows for assessment of both increased and decreased survival in response to exposure for treatment, in this case by berry extracts.

### 2.3. Effect of In Vitro Digested Berry Extracts on Yeast Survival

After establishing suitable stress conditions, *S. cerevisiae* SKQ2n was cultured in a synthetic medium supplemented with different berry anthocyanin extracts (bilberry, black currant, strawberry, and raspberry) to assess cell viability during stress. In addition to using various berries, anthocyanin extracts with and without exposure to simulated GI-digestion were compared in terms of cell survival (Figure 3).

All nondigested berry extracts, representing the mixture of anthocyanin as they would be when entering the mouth, decreased the yeast’s survival capacity. However, all in vitro GI-digested extracts, representing anthocyanins in forms would be present in the small intestine, enhanced yeast survival. The highest increase in survival capacity was observed in cells cultured with digested strawberry extracts (Figure 3D); approximately 70% increase in viability after 10 min of oxidative stress when compared to the control. Digested extracts from raspberry, blackcurrant, and bilberry increased the survival rate by approximately 62, 53 and 47%, respectively, in cells treated with H_2_O_2_ for 15 min (Figure 3A–C) Thus, GI-digested anthocyanins roughly doubled the resistance of yeast cells against severe oxidative stress.

Overall, nondigested berry extracts reduced the survival of oxidatively stressed yeast cells. However, the reduction was the smallest with strawberry extracts (Figure 3A–D). To verify that the protective effect from the digested extracts was due to the berry compounds alone and not to the digestion physicochemical mixture, an experiment with the digestion mix without added berry extracts (background) was also conducted. This resulted in a survival curve that closely followed the control (i.e., no extract, no digestion reagents) suggesting that the observed cell protection resulted solely from exposure to berry compounds (Figure 4).

### 2.4. Protective Effect Remains after the Removal of Digested Extracts

To address whether the observed protection was a chemical neutralization of the H_2_O_2_ effect outside the cells, or if the berry compounds induced a biological response inside the yeast cells leading to a phenotype with enhanced stress resistance, two more experiments were performed: (1) cultivation of yeasts in the presence of digested extract followed by removal of the extract before the stress assay and (2) cultivation of yeasts without extract followed by addition of the digested extract before the stress assay. For these experiments, bilberry was selected, since prior work suggested that bilberry anthocyanins may reduce oxidative damage as well as inflammation in humans [25,26,27,28,29]. Results showed that the cells grown in yeast medium plus digested extract displayed a significantly higher stress resistance (Figure 5A) compared to the control. On the other hand, no increase in survival was observed when cultures were exposed to berry extracts only during H_2_O_2_ stress exposure (Figure 5B). These results suggest that the stress-resistant phenotype was formed only when the yeast cells were propagating and adapting in the presence of the digested berry compounds.

## 3. Discussion

Phenolic compounds have been reported to induce health benefits in humans [30]. Some studies show that these benefits may originate from the flavonoid anthocyanins or other, simpler phenolic acids abundant in berry fruits [31,32]. Data suggests that (poly)phenols, including anthocyanins, can eventually be used to treat certain diseases such as chronic and neurodegenerative conditions and cardiovascular disease (CVD) [33,34]. The health benefits of anthocyanins are commonly suggested to originate from their antioxidant and anti-inflammatory activities related to their chemical structure [35]. However, epigenetic activities have also been associated with flavonoids, which seem to induce changes in gene expression protecting, for example, against cancer [36]. The mechanisms of cell protection by polyphenols are not fully understood, and neither is the relative degree of protection by intact anthocyanins versus their derivatives resulting from GI-digestion. Thus, the present study aimed at investigating the effect of digested anthocyanins on cells using the unicellular model-organism *S. cerevisiae*. We focused on comparing cellular protection from intact berry extracts against the same extracts exposed for GI-digestion to attain our aim. Overall, the results showed that the digested extracts from bilberry, blackcurrant, strawberry and raspberry had a significant protective effect on yeast cells. These findings agree with previous studies reporting protective effects of digested (poly)phenol-rich foods against oxidative injury in yeast and human cell models [37,38,39]. Likely these effects were dependent on chemical alterations of anthocyanins caused by the digestion process.

For nondigested extracts, the impact on yeast cells was the opposite of that observed for digested extracts, i.e., a decreased stress resistance was observed suggesting a disturbed defence against oxidative stress. This suggestion agrees with the findings reported by Wei et al. [36]. They showed that in human cells the phenolic propyl gallate, synthesized by the condensation of gallic acid and propanol, might, in certain conditions, increase the levels of reactive oxygen species (ROS) and/or promote depletion of protective enzymes or glutathione. Moreover, high (poly)phenol levels can inhibit enzymes such as peroxidases and catalases [40,41]. It is also well known that (poly)phenols from berries can exhibit antimicrobial activity [42]. In our study, the yeast cells cultivated in the presence of nondigested anthocyanins grew at an equal rate as controls (similar OD at harvest time nine hours), suggesting no cell-toxicity at nonstress conditions (data not shown). However, subsequent stress reduced survival in yeasts grown on nondigested extracts.

The well-known antioxidative effect of (poly)phenols might suggest a direct interaction with H_2_O_2_ leading to reduced oxidative attack, explaining the observed protection by digested extracts. However, the protective effect persisted in the cells when the extracts were removed before the stress assay. Thus, these results strongly suggest a biological effect rather than a direct chemical neutralization of H_2_O_2_. For example, a change in phenotype during growth is induced by digested anthocyanins. The opposite experiment further strengthened this observation: the absence of anthocyanins during growth and presence during the stress assay showed no protective effect. Our results suggest that the yeast cells somehow changed their phenotype to a more resistant state when growing in the presence of the anthocyanin derivatives. This state remained after removing the compounds. Thus, it is possible that the cells sensed and were able to take up these anthocyanin derivative molecules. Inside the cytoplasm, the anthocyanin derivatives somehow affected the defense state against oxidative stress. Either this happened by changes in gene expression or by altering the activity of the existing pool of enzymes. However, we do not know which alterations in gene expression or enzyme activities occurred due to the exposure of yeast cells to berry anthocyanin derivatives. It is also important to consider that the in vitro GI-digestion approach generates transformed phenolic compounds from berries to mimic the derivatives reaching the small intestine [43]. Naturally, the human gut complexity is more significant due to the GI-biota, which was absent in our model.

Although anthocyanin recovery was in general very low after in vitro digestion, the digested extracts of strawberry showed the highest protective effect compared to the extracts from other berries. In a previous study we also showed a low recovery of cyanidin-3*-O-*glucoside and malvidin-3*-O-*glucoside after in vitro digestion of strawberry and blueberry, respectively [44]. Garcia et al. [21] also reported that only 19.3% of total phenols were recovered after in vitro GI digestion of raspberry, while most phytochemicals present in the original extracts suffered substantial modification after digestion. Furthermore, anthocyanins are amongst the most instable polyphenols, with studies showing that after in vitro digestion red raspberries were almost depleted in anthocyanins, particularly pelargonidin, compared to the nondigested extracts [22]. Likely, the metabolites formed after in vitro digestion of anthocyanins also played a role in protecting yeast cells from stress caused by H_2_O_2_. An alternative explanation to our results is that a higher dose of berry extracts/anthocyanins might be toxic to the yeast cells whereas a lower (as after digestion) would be protective against stress (hormesis). 

Strawberries were also the least diverse in the type of anthocyanins identified and had the lowest total anthocyanin content compared to the other berries, particularly bilberry. As in our study, others have also reported that total anthocyanin content for bilberry was significantly higher than for strawberry (33 and 8 mg g^−1^, respectively) [45,46]. Moreover, the anthocyanin profile of strawberry is markedly different from bilberry, blackcurrant, or raspberry. Pelargonidin-3*-O-*glucoside was reported as the most abundant anthocyanin in strawberries, representing around 54% of the total anthocyanin content, followed by other types of pelargonidin glycosides and aglycones [46]. In our study, cyanidin-3*-O-*rutinoside was present in a higher amount in strawberries compared to other compounds, whereas pelargonidin-3*-O-*glucoside was not identified. However, the nonglycosylated form, pelargonidin, was identified, suggesting that the sugar moiety might have been hydrolyzed during sample processing. Like the results obtained in our study, others also reported that bilberry has a higher diversity of anthocyanins, among which delphinidin and cyanidin glucosides are most abundant. Blackcurrant is rich in delphinidin and cyanidin rutinosides [47], and in raspberry, cyanidin-3*-O-*sophoroside was reported as the main anthocyanin [48].

Overall, in the presence of anthocyanins and its metabolites, particularly from digested strawberry, a significant increase in cell survival was observed, suggesting that not only the type and amount of anthocyanins present in yeast cultures during cell growth is important, but that anthocyanins need to undergo GI structural modifications to induce cell defence. Further studies are required to identify the major differences in polyphenol profiles and its derivatives after in vitro digestion of different berry fruits and their bioactive roles against cell stress.

## 4. Materials and Methods

### 4.1. Materials

Strawberries (*Fragaria ananassa*), of the variety Malwina, and blackcurrant (*Ribes nigrum*) of the variety Ben Tron were collected in July 2016, in Dingle and Fagerfjäll, Sweden, respectively. Raspberries (*Rubus idaeus*), of the variety Glen Ample, were harvested in June 2016 in Norway. Bilberry powder, obtained by freeze-drying and grinding whole bilberries (*Vaccinius myrtillus*), a wild variety of blueberry, was provided by Immun Skellefteå, Sweden (www.immun.se). All fruit samples were freeze-dried, milled and stored at −80 °C before analysis.

Anthocyanin standards: cyanidin-3*-O-*arabinose, cyanidin-3*-O-*galactoside, cyanidin-3*-O-*glucoside, cyanidin-3*-O-*rutinoside, delphinidin-3*-O-*galactoside, delphinidin-3*-O-*glucoside, delphinidin-3*-O-*rutinoside, malvidin-3*-O-*galactoside, malvidin-3*-O-*glucoside, pelargonidin-3*-O-*glucoside and cyanidin, delphinidin, malvidin, pelargonidin, peonidin aglycones were all purchased from Extrasynthese (Genay, Rhône, France).

Alfa-amylase was obtained from porcine pancreas (Sigma-Aldrich, St. Louis, MO, USA, cat no. A3176), porcine pepsin, (Sigma Aldrich, St. Louis, MO, USA, cat no. P6887), porcine pancreatin (Sigma Aldrich, St. Louis, MO, USA, cat no. P7545) and porcine bile (Sigma Aldrich, St. Louis, MO, USA, cat no. B8631).

### 4.2. In Vitro Digestion of Berry Samples

The in vitro method to simulate digestion was based on the protocol described by Minekus et al. [49], with some modifications. The method consists of three sequential steps: an initial step simulating the oral phase followed by an intermediate step with pepsin/HCl mimicking conditions in the stomach, and a third step with bile salts/pancreatin simulating conditions in the small intestine. The simulated fluids, simulated salivary fluid (SSF), simulated gastric fluid (SGF) and simulated intestinal fluid (SIF), were prepared according to Minekus et al. [49]. Briefly, in the oral step, 3.0 g of dried berry sample (in triplicate) were mixed in a tube containing 5 mL of simulated salivary fluid (SSF), and 2.0 g of this mixture was transferred to a new tube and mixed with 5 mL of alfa-amylase solution (porcine, 112 U/mL). For calculation of anthocyanin content, an aliquot (~1 g) of each sample was removed and used for dry matter determination, which was similar in all samples (~10%). The tube was vortexed for 15 s following incubation for 2 min at 37 °C on a rotary shaking plate (250 rpm). In the gastric phase, 5 mL of pepsin solution (5000 U/mL), prepared by dissolving pepsin in simulated gastric fluid (SGF) was added to the previous tube, and the pH was adjusted to 3.0 using HCl (1 M). The tube was incubated for 2 h at 37 °C on a rotatory shaking plate (250 rpm). In the intestinal phase, 5 mL of bile solution (45 µmol/mL) and 5 mL of pancreatin solution (450 U/mL), both prepared by dissolving the solutes in simulated intestinal fluid (SIF), were added to each tube and the pH was adjusted to 7.0 using NaOH (1 M). The tube was incubated at 37 °C on a rotatory shaking plate (250 rpm) for 2 h. The samples were frozen (−80 °C), freeze-dried and stored at −20 °C until extraction of anthocyanins.

### 4.3. Extraction of Anthocyanins

Anthocyanins were extracted from digested and nondigested samples according to the methods of Bunea et al. [10] and You et al. [50], with some modifications. The freeze-dried samples (0.200 g ± 0.015 and 1.00 g ± 0.05, respectively) were mixed by vortexing for 30 s with 5 mL of methanol with 0.3% HCl (*v*/*v*) in a glass tube with a screw cap. All tubes were blanketed with nitrogen gas, sealed and placed in the dark at 4 °C for 18 h. After sonication for 15 min at 20 °C, 37 kHz (S15 Elma Sonicator, Elma Schmdbauwer, GmbH, Singen, Germany), the samples were centrifuged at 2000× *g* for 10 min, and the supernatant fluid was collected. Re-extraction was carried out by adding 5 mL of acidified methanol to the pellet, followed by vortexing, centrifugation and finally pooling the supernatants from each extraction. The supernatants (10 mL) were centrifuged at 4000× *g* for 15 min. The solvent was evaporated from the supernatants, and the solid residues were then resolubilized in YNB media and stored at −20 °C until analysis.

### 4.4. HPLC-UV/VIS Analysis of Anthocyanins

Anthocyanin profiles of digested and nondigested bilberry, blackcurrant, raspberry, and strawberry extracts were determined using high-performance liquid chromatography (HPLC). The HPLC system consisted of a quaternary gradient pump (Jasco PU-2089 Plus; Jasco Inc., Easton, MD, USA), a cooled (8 °C) autosampler (JascoAS-2057 Plus, Jasco Inc., Easton, MD, USA), and a UV detector operating at 520 nm (Shimadzu SPD-10A UV-Vis detector; Shimadzu Corp., Kyoto, Japan). Mobile phases consisted of 5% aqueous formic acid (A) and methanol (B). A linear gradient at a flow rate of 0.5 mL/min was used with an increase from 13 to 25% of eluent B over 25 min, followed by an increase to 50% of eluent B over 2 min then isocratic elution at 50% eluent B for 5 min. Samples were then returned to initial conditions and the column reconditioned at 13% B for 5 min prior to injection of the next sample. A Jasco ChromNAv software was used to control the HPLC system and for data processing. Separation of the individual anthocyanins was achieved with a Phenomenex Luna C18 column (250 mm × 3.0 mm, 3 μm) at 40 °C.

### 4.5. Yeast Strain and Growth Conditions

The diploid yeast strain *Saccharomyces cerevisiae* SKQ2n (ATCC 44827; a/alpha; ade1/+; ade2; +/his1) was used in all experiments. This strain has been previously used as a reference/model strain in fundamental studies of the osmotic stress response [51,52]. In this study, yeast cells were stored at −80 °C in 20% glycerol until use. The cells were then cultivated in a medium (sterilized by filtration) containing 0.67% (*wt/vol*) yeast nitrogen base (YNB, Difco Laboratories, Detroit, MI, USA) with 5% ammonium sulfate, supplemented with 2% glucose, and grown at 30 °C in 10 mL tubes in a LabRollerTM Rotator at 30 rpm and maintained on plates with YPD agar (per L: 1−0 g of yeast extract, 20 g of peptone, 20 g of d-glucose and 20 g of agar). Optical densities were measured at 600 nm (OD600) using a ULTROSPEC 10 cell density meter (Amersham Biosciences, Little Chalfont, UK).

### 4.6. Assessment of the Optimal Concentration of H_2_O_2_ for Oxidative Stress

To determine the working concentrations of H_2_O_2_ and, consequently, to evaluate the ability of berry extracts to protect against oxidative stress, a preliminary study was conducted as follows. A preculture was grown in YNB (5 mL) supplemented with 2% glucose for approximately 9 h (20 °C) until an OD600 of 6 was obtained. The cells were harvested by centrifugation at 4000× *g* for 5 min. The pellet was resuspended in 0.9% NaCl and adjusted to an OD600 of 1.0 in a 10 mL culture tube and subsequently diluted 10 times. Oxidative stress was induced by dividing the cellular suspension into 10 tubes and adding H_2_O_2_ (30% *w/w*) to a concentration of 0.15, 0.3, 0.45, 0.6, 0.75, 0.9, 1.0, 1.2, and 1.4 M to a final volume of 1 mL for each sample. The samples were incubated at 20 °C with shaking at 180 rpm for 10 min and subsequently immediately diluted 10 times in 10 mL tubes containing saline to stop the stress exposure. Analyses of viable cells were performed by making serial dilutions with 0.9% NaCl in the range of 101–104 for each sample and spreading 10 µL drops of the 102–104 dilutions onto YPD plates pentaplicates. The plates were incubated at 30 °C for two days, and cell viability was measured as CFU (colony forming units). The relative survival rate of the yeast cells was calculated as a percentage by dividing the number of colonies, CFU/mL after exposure to H_2_O_2_, by the number of CFU/mL in untreated cells (control), which was considered as 100%.

### 4.7. Assessment of the Protective Effect of Berry Extracts against Oxidative Stress

A preculture was grown and harvested as described above. The pellet was suspended and adjusted to OD_600_ of 0.2 in culture tubes containing YNB supplemented with 2% glucose, chloramphenicol (200 mg L^−1^), and digested or nondigested berry extracts. The concentration of berry extracts was set at 10 mg mL^−1^, reported as nontoxic to *S. cerevisiae* [53,54]. The tubes were incubated at 30 °C in a rotator at 30 rpm for 9 h. As controls, tubes containing the yeast cells without berry extracts or reagents used during the in vitro digestion experiments were cultivated using the same conditions as described above. Since phenolic compounds may act in synergy with the digested berry extracts, the yeast cells were also cultivated in YNB supplemented with 2% glucose, chloramphenicol (200 mg L^−1^), and the extracts from in vitro digested berry.

The oxidative stress assay with H_2_O_2_ was made in triplicates (three independent yeast suspensions) for each experimental mix as described above, at room temperature (20 °C) (180 rpm, 45 min). At intervals, samples (100 μL) were removed from each cellular suspension every 5 min and immediately diluted in 900 µL saline using 1.5 mL sterile microcentrifuge tubes to stop the stress exposure. In parallel, the cellular suspensions were diluted, spread onto YPD plates, analysed for the number of colonies, and survival was calculated according to the procedure described above.

To further evaluate whether the berry-induced effect was a direct physicochemical effect or required a time dependent biological response, bilberry was selected for two additional experiments: (I) samples cultured with bilberry extract were washed with 0.9% NaCl to remove the extracts from the cells just before exposure to H_2_O_2_ and (II) yeast cells were cultivated without bilberry extract using the same growth conditions described previously, with the addition of the bilberry extract to the cultured tubes 5 min before H_2_O_2_ exposure.

### 4.8. Statistical Analyses

The Statistical Analysis System computer package (SAS Institute, Inc., Cary, NC, USA, 2004) was used to analyse data. The data were treated by two-way analysis of variance (ANOVA) with nondigested and digested treatments as main effects. Significant differences between treatments were detected using Tukey’s Studentized Range (HSD) test at the 5% level of significance. The results of the statistical analyses are given in the Supplementary Material (Appendix A).

## 5. Conclusions

In conclusion, this study shows that in vitro GI-digested extracts from anthocyanin-rich berries can protect yeast cells stressed with H_2_O_2_, while nondigested extracts, in contrast, increased their sensitivity to oxidative stress. The results suggest a biological activity of digested anthocyanins and their derivatives on cell protection against oxidative stress. Further research using a similar approach should be conducted using human cells such as Caco-2. The highest level of cell protection was found when extracts from strawberry were used, the richest source of pelargonidin glycosides in our study. However, further studies are needed to elucidate the mechanisms behind our findings and investigate, for instance, the expression of genes related to oxidative stress

## Figures and Tables

**Figure 1 molecules-26-01057-f001:**
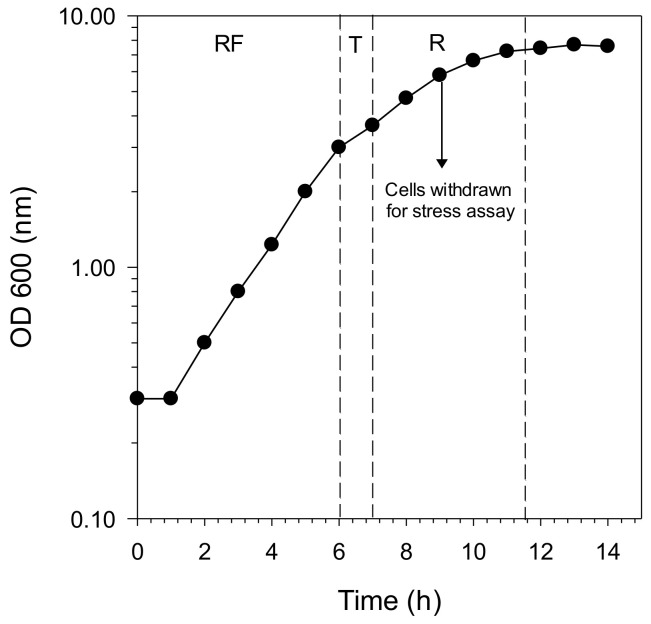
Growth curve of the *Saccharomyces cerevisiae* SKQ2n cultured in synthetic medium YNB (Yeast Nitrogen Base) with 2% glucose as the sole carbon and energy source and monitored by optical density (OD). Dotted lines represent transitions from one physiological state to another. RF: respiro-fermentative phase; T: transition phase and R: respiratory phase.

**Figure 2 molecules-26-01057-f002:**
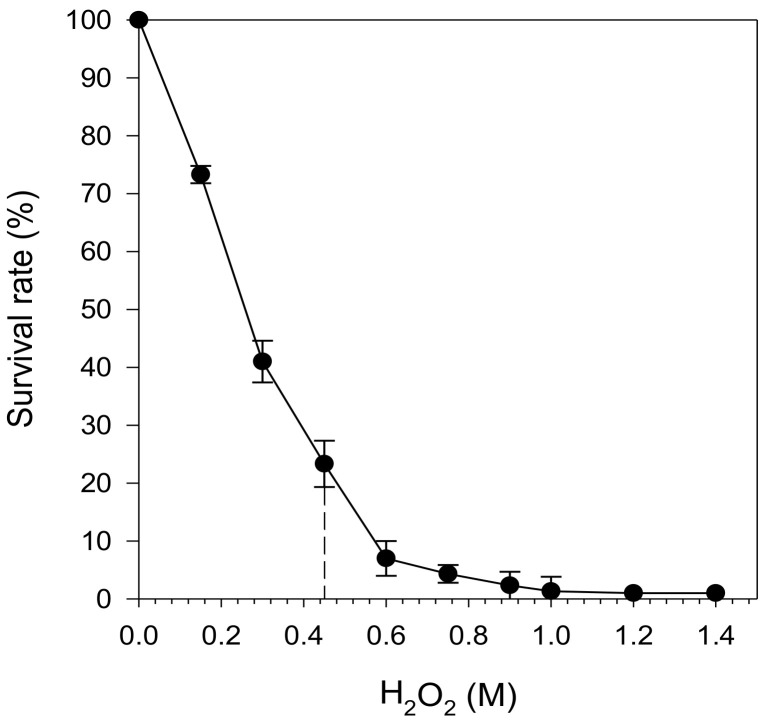
Survival of the yeast strain SKQ2n (expressed as percentage of initial cell concentration) as a function of 10 min exposure of different H_2_O_2_ concentrations. A concentration of 0.45 M (dashed line) was selected to perform the stress experiments.

**Figure 3 molecules-26-01057-f003:**
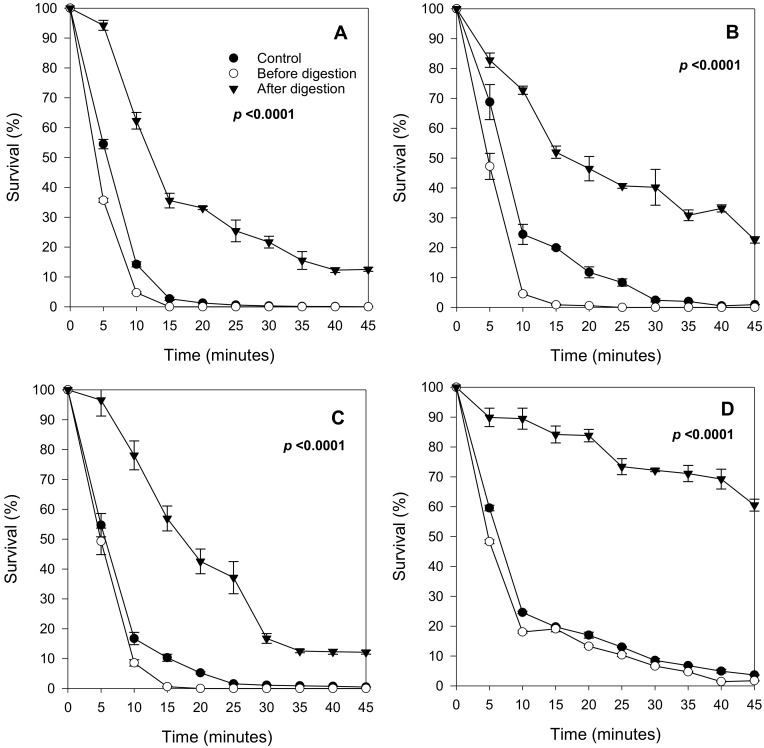
Survival curves of *S. cerevisiae* SKQ2n treated with H_2_O_2_ (0.45 M) cultured in YNB with 2% glucose without berry extracts (Control), with berry extracts before in vitro digestion (before) and after in vitro digestion (After). (**A**) = bilberry, (**B**) = black currant, (**C**) = raspberry and (**D**) = strawberry. Significant differences between treatments were detected using Tukey’s Studentized Range (HSD) test at the 5% level of significance (see also suplementary materials Appendix A). Data points are averages of replicated samples from three separate (independent) stress assays (±SE).

**Figure 4 molecules-26-01057-f004:**
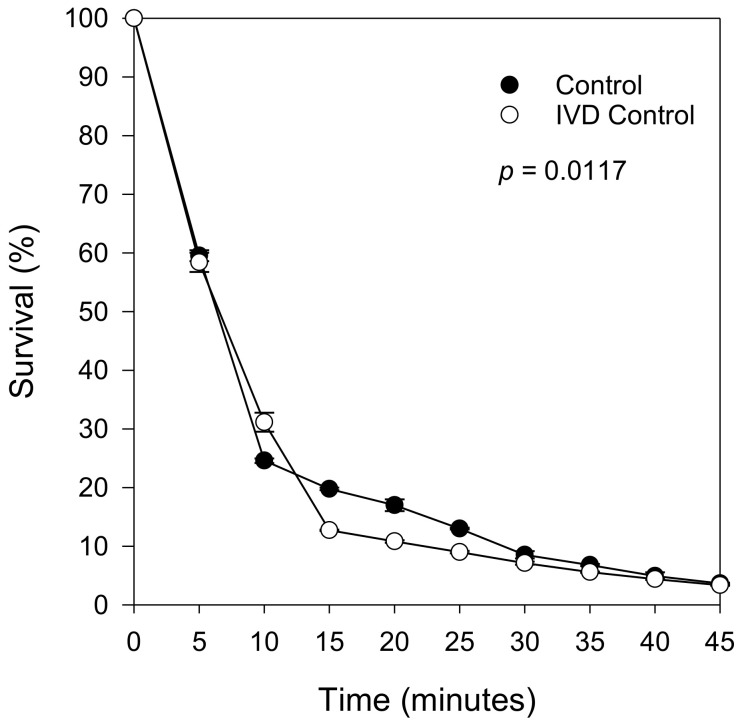
Survival curves during H_2_O_2_ stress of *S. cerevisiae* SKQ2n cultured in YNB with 2% glucose without berry extracts (control) and after in vitro digestion without berry extracts (IVD blank; only in vitro digestion background chemicals, without berry extracts). Significant differences between treatments were detected using Tukey’s Studentized Range (HSD) test at the 5% level of significance. Data points are averages of replicated samples from three separate (independent) stress assays (±SE).

**Figure 5 molecules-26-01057-f005:**
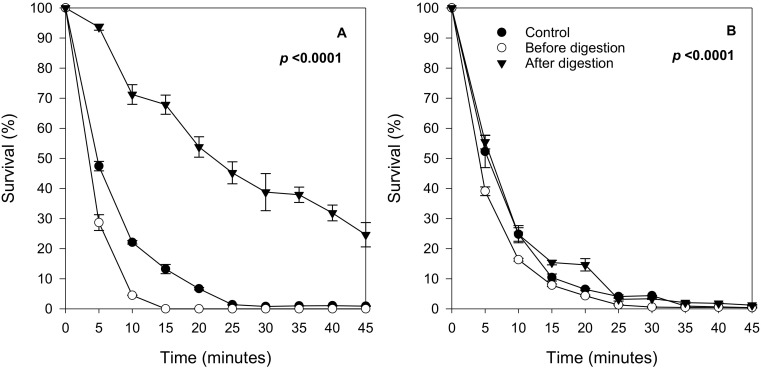
Survival curve of the yeast strain SKQ2n cultured with bilberry. (**A**) Cells cultured with bilberry extracts but washed before the H_2_O_2_ treatment; (**B**) cultured without bilberry extracts but mixed with bilberry extracts 5 min before H_2_O_2_ treatment. Cultures without bilberry extract (Control), with nondigested bilberry extracts (Before) and after in vitro digestion (After). Significant differences between treatments were detected using Tukey’s Studentized Range (HSD) test at the 5% level of significance. Data points are averages of replicated samples from three separate (independent) stress assays (±SE).

**Table 1 molecules-26-01057-t001:** Anthocyanin profiles of digested and nondigested bilberry, blackcurrant, raspberry and strawberry.

	Nondigested (mg g^−1^ DW ^d^)	SE ^a^	Digested (mg g^−1^ DW)	SE	*p* Value (α = 0.05) ^b^
**Bilberry**					
Cyanidin-3*-O-*arabinose	1.43	0.07	0.07	0.01	0.003
Cyanidin-3*-O-*galactoside	0.78	0.05	0.06	0.00	0.005
Cyanidin-3*-O-*glucoside	1.29	0.06	0.10	0.01	0.003
Cyanidin-3*-O-*rutinoside	0.51	0.03	0.04	0.00	0.004
Delphinidin-3*-O-*galactoside	1.07	0.05	0.05	0.01	0.003
Delphinidin-3*-O-*glucoside	1.91	0.10	0.04	0.00	0.003
Delphinidin-3*-O-*rutinoside	1.29	0.07	0.03	0.00	0.003
Malvidin-3*-O-*galactoside	0.43	0.02	0.05	0.00	0.005
Malvidin-3*-O-*glucoside	1.32	0.06	0.14	0.01	0.003
Pelargonidin-3*-O-*glucoside	0.15	0.02	0.02	0.00	0.022
Cyanidin aglycone	0.23	0.01	ND ^c^		
Delphinidin aglycone	0.19	0.01	0.01	0.00	0.001
Malvidin aglycone	0.08	0.00	ND		
Pelargonidin aglycone	1.04	0.05	0.11	0.01	0.003
Peonidin aglycone	0.07	0.00	ND		
Sum of anthocyanins	11.78		0.71		
**Blackcurrant**					
Cyanidin-3*-O-*arabinose	0.067	0.004	ND		
Cyanidin-3*-O-*galactoside	2.505	0.250	0.261	0.021	0.012
Cyanidin-3*-O-*glucoside	0.322	0.036	0.045	0.004	0.016
Cyanidin-3*-O-*rutinoside	1.258	0.128	0.182	0.013	0.013
Delphinidin-3*-O-*glucoside	0.739	0.074	0.074	0.007	0.012
Malvidin-3*-O-*galactoside	0.028	0.016	0.005	0.002	0.221
Malvidin-3*-O-*glucoside	0.018	0.018	ND		
Pelargonidin-3*-O-*glucoside	ND		0.003	0.003	
Delphinidin aglycone	ND		0.002		
Malvidin aglycone	ND		0.001	0.001	
Pelargonidin aglycone	0.026	0.013	0.004	0.000	0.237
Peonidin aglycone	0.063	0.001	ND		
Sum of anthocyanins	5.03		0.58		
**Raspberry**					
Cyanidin-3*-O-*galactoside	0.114	0.007	0.009	0.001	0.005
Cyanidin-3*-O-*glucoside	0.129	0.003	0.014	0.001	0.001
Cyanidin*-O-*rutinoside	0.026	0.001	0.003	0.000	0.003
Delphinidin-3*-O-*glucoside	0.623	0.020	0.050	0.005	0.002
Delphinidin-3*-O-*rutinoside	ND		0.002	0.000	
Sum of anthocyanins	0.89		0.08		
**Strawberry**					
Cyanidin-3*-O-*glucoside	0.034	0.001	0.003	0.001	0.001
Cyanidin-3*-O-*rutinoside	0.656	0.003	0.087	0.015	0.001
Cyanidin aglycone	ND		0.005	0.001	
Pelargonidin aglycone	0.049	0.000	0.007	0.001	0.001
Sum of anthocyanins	0.74		0.10		

^a^ SE = Standard error. ^b^ Significant difference between treatments were detected using the Tukey’s Studentized Range (HSD) at the 5% level of significance. ^c^ ND = Not-detected. ^d^ DW = Dry weight.

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
