# Peer review of "Extracts of Digested Berries Increase the Survival of Saccharomyces cerevisiae during H2O2 Induced Oxidative Stress"

_molecules, 2021, doi:10.3390/molecules26041057_

Round 1
Reviewer 1 Report
General Impression
In order to improve our understanding of the presumed cytoprotective effects of plant extract, the authors describe the results of a study on the antioxidant effects of a methanolic extract of red berries on baker’s yeast, Saccharomyces cerevisiae. To mimic the effects human digestion, the authors subject the plant extracts to an established in vitro digestion procedure. Anthocyanin contents of extracts pre- and post-digestion are determined by HPLC, and the effects of plant extracts on yeast survival and H2O2 resistance are examined by survival assay. The authors find that digestion changes the extracts’ anthocyanin concentrations, with a marked reduction in overall content. Subsequent experiments show that the plant extract pre-digestion has cytotoxic effects, whereas the digested extract induces protection to oxidative stress. The authors conclude that the digestive process transforms anthocyanins in plant extracts into cytoprotective molecules. While the experiments are solid, the conclusions are not valid without additional control and I suggest the manuscript be returned to the authors for additional work. As presented, various alternative explanations of the findings are plausible, which greatly diminishes the value of the manuscript to the reader.
Major points of criticism
A. Limitations of analysis - Composition of Extract?
- The authors examine the biological effects of methanolic extracts of bilberry, blackcurrant, raspberry and strawberry. It is assumed that anthocyanins are the active components in the extract, and a HPLC analysis of anthocyanin content is provided. However, methanolic extracts are known to contain many hundreds of other bioactive molecules, which makes the method a standard protocol for extracting medicinal plant tissues. The study would benefit from a broader analysis of the extracts’ contents.
B. Unclear dose-effect: Is there hormesis?
- The authors subject the extracts to in vitro digestion and find that the anthocyanin content is reduced by the procedure. This suggests that the bioactivity of the extract is reduced by the same amount.
- The authors characterize the effects of the plant extract at a single dose. This is a critical flaw in the argument. There is often a hormetic effect in antioxidant action of phytochemicals, with very high concentrations of the compound having a negative effect on survival (see e.g. recent review by Martel et al., Trends in Endocrinology and Metabolism 30(6), 335-346). A dose-response curve is required for a solid understanding of the extracts’ effects.
- The authors observe that digested extracts have antioxidant effects while undigested extracts are cytotoxic. Having established that digestion reduces anthocyanin content of extracts, this finding suggests hormesis rather than an unspecified modification of biomolecules.
C. Unclear biological effect of extract
- Baker’s yeast has certain advantages for the examination of the eukaryotic stress response. However, the unspecific nature of the response poses a major problem for the interpretation of experiments such as the ones presented here. As Gasch et al. pointed out in their early work (e.g. Mol Biol Cell 2000), yeast activates a generic environmental stress response that does not necessarily correlate with the actual stressor: The finding that the extract induces protection against oxidative stress does not mean that the extract is in itself an oxidative stressor or protector. This needs to be discussed or addressed experimentally (e.g. does one induce heat shock resistance by pre-treatment with plant extract?).
Minor point
The graphic representation of yeast survival is unusual. Please re-draw using to more common log-plot.
Reviewer 2 Report
Dear Authors,
I have read your work with pleasure. The research idea and scientific approach of this work are without major problems. Unfortunately I found a few problems that need to be corrected. Namely:
- What was the plant material used for the research, what part of the plants (roots, fruits, aerial parts, flowers)? From introduction section I guess it was fruits, but it must be descibed in the experimental section.
- Was the plant material harvested in the same growing season at the same time? When? On september, october? The harvesting time can influence on the level of secondary metabolites.
- In the Table 1.: mg g-1 of fresh or dry weight? Be carefull! They are not the same values.
- The anthocyanin prophiles and contents in bilberry, blackcurrant, raspberry and strawberry fruits (? I don't know.) were measured by HPLC methods. Tell me please (it must be written exactly in the experimental section), based on what were these chemicals identified (retention times? UV-spectra? If yes, there is not enough data for anthocyanin identification) and quanticated (on the basis of what? Callibration curves? What the concentration ranges?)? Did you use the standard compounds of anthocyanin (own isolated? If yes, what the method of separation was used? If the standards were bought, then from where (what producer?)? )
- All HPLC chromatograms of all separated extracts must be included in the result section, each in the same scale.
I hope that the above remarks will not discourage you from improving this work, especially as the topic is very interesting.
Best regards - reviewer.
Reviewer 3 Report
In this study, Andlid and colleagues set out to investigate the effect of simulated digestion (using in vitro gastrointestinal conditions) on the anthocyanin profile of four berry extracts. Moreover, they assessed the protective effects of digested and undigested extracts on yeast cells challenged with hydrogen peroxide. The main motivation behind the study is that, although much is known about the beneficial effects of anthocyanins on biological systems, little is known about their actual effect in the organisms after gastrointestinal digestion. Expectedly, simulated digestions greatly reduced the level of anthocyanins. Moreover, digested extracted exerted a protective effect against hydrogen peroxide-induced stress, whereas non-digested extract had no effect. The methodology is scientifically sound, and the experiments removing the extract prior to H2O2 exposure or adding extract right before adding H2O2 were well designed. Disappointingly, authors have not explored the wide variety of available yeast strains to explore the molecular pathways involved in their system. For example, the use of mutant strains, such as yap1∆, could shed some light on the actual mechanism. Moreover, there is no mention to the results of statistical analyses between digested, non-digested, and control in the text or in the figures. Please see my concerns numbered below:
- Lines 92, 95, and elsewhere: Please use “significant” to refer to those results where statistical significance was calculated. The study does not provide results of statistical analyses between berries.
- Lines 131–132: From table 1, only one new compound (cyanidin aglycone) was detected in the digested extract compared with the non-digested strawberry extract. Please check.
- Line 185: Please check this statement. Strawberry is shown in panel D, not C.
- Lines 188–189: The letters do not agree with those in figure legend. Please check.
- Please provide the results of the statistical test described in section 4.8 to compare digested, non-digested and control yeast cells. Moreover, please include symbols in the figure to denote statistically significant differences. Otherwise, the statements about decrease or increase in cell survival are unjustified. The same is true for the comparison between IVD control versus control; the results of the statistical analyses must be given.
- Please explain to the readers why authors chose to use bilberry extracts in the experiments shown in figure 5, whereas strawberry had the highest effect on cell survival. What was the rationale? Would not it be a better approach to test the four extracts in this set of experiments?
- Lines 266–284: Consider using the following reference to support the discussion on the effects of polyphenols on antioxidant systems: 10.1016/j.freeradbiomed.2013.05.045
- The apparent paradox of lower anthocyanin levels exerting higher beneficial effects fits a key concept in biology: hormesis. Please consider re-framing part of the discussion in the light of the hormesis concept. See these refs: 10.15698/mic2014.05.145 ; 10.1038/s41514-017-0013-z ; 10.1038/sj.embor.7400222.
- Line 335: Please describe the composition of the simulated salivary fluid.
- Line 336: Here, authors report that a sample was retained form moisture analysis. Where is the result?
- Lines 337–342: Please provide the concentration (e.g. mg/mL, IU/mL, etc.) of the enzymes and the “bile solution” used to simulate GI digestion. Preferably, provide concentrations in mass per volume so that the reader can have a clear view of possible diluting effects of the added digesting agents. In other words, some of the “loss” of anthocyanin content might be due to the addition of external mass (from enzymes, salts, etc.) that is not removed during drying.
- Line 398: Please provide the exact temperature or temperature range in which cells were incubated. “Room temperature” is not informative.
- Line 425: The text describes that “berry extracts” were tested, but only results for bilberry extracts are shown in figure; there are no results for strawberry, raspberry and blackcurrant. Considering that strawberry extracts had the most prominent results, it is not clear why authors chose to use bilberry in this set of assays.
- The concentration of hydrogen peroxide used for optimization and experiments seems too high; it is 1.4 molar. Most studies work within the 0.1–10 mM range (e.g., 10.1590/S0100-879X2004000200001 ; 10.1016/0014-5793%2895%2900603-7 ; 10.1128/jb.174.20.6678-6681.1992 ; 10.3389/fmicb.2018.01933). Here in the lab, for example, we have successfully inhibited the growth of BY4741 yeast cells using 1.0 mM (unpublished data). Therefore, I invite authors to briefly discuss this huge discrepancy between their results and those available in the literature in the discussion.
Minor issues
- Lines 31–32 and elsewhere: Use italics for species names.
Round 2
Reviewer 1 Report
The authors have opted to address the concerns in the text rather than report additional experiments. This is permissible, but it reduces the scientific value of the the paper. The sentence addressing the possibility of a simple hormetic effect needs editing ("An alternative explanation to our results is hormesis. In our work that would mean that the lower dose of anthocyanins was beneficial whereas the higher was negative").
Reviewer 2 Report
Dear Authors,
Thank you for placing the chromatograms and the detailed description of the HPLC method. Although the retention times of the standard substances are slightly lower than the analytes in the extracts, this is often the case and is acceptable. This may be due to the high viscosity of the extract samples. I know, that increasing the column temperature could compensate for this, but you already used 40 degrees of Celsius. Further increasing the temperature could destroy the measured metabolites and the column. If you correct other reviewer's comments in the text and in the margins, the work can be accepted.